# On the Proton-Bound Noble Gas Dimers (Ng-H-Ng)^+^ and (Ng-H-Ng’)^+^ (Ng, Ng’ = He-Xe): Relationships between Structure, Stability, and Bonding Character

**DOI:** 10.3390/molecules26051305

**Published:** 2021-02-28

**Authors:** Stefano Borocci, Felice Grandinetti, Nico Sanna

**Affiliations:** 1Dipartimento per la Innovazione nei Sistemi Biologici, Agroalimentari e Forestali (DIBAF), Università della Tuscia, L.go dell’Università, s.n.c., 01100 Viterbo, Italy; borocci@unitus.it (S.B.); n.sanna@unitus.it (N.S.); 2Istituto per i Sistemi Biologici del CNR, Via Salaria, Km 29.500, 00015 Monterotondo, Italy; 3Istituto per la Scienza e Tecnologia dei Plasmi del CNR (ISTP), Via Amendola 122/D, 70126 Bari, Italy

**Keywords:** bonding analysis, chemical bond, noble-gas chemistry, noble-gas ions, structure and stability

## Abstract

The structure, stability, and bonding character of fifteen (Ng-H-Ng)^+^ and (Ng-H-Ng’)^+^ (Ng, Ng’ = He-Xe) compounds were explored by theoretical calculations performed at the coupled cluster level of theory. The nature of the stabilizing interactions was, in particular, assayed using a method recently proposed by the authors to classify the chemical bonds involving the noble-gas atoms. The bond distances and dissociation energies of the investigated ions fall in rather large intervals, and follow regular periodic trends, clearly referable to the difference between the proton affinity (PA) of the various Ng and Ng’. These variations are nicely correlated with the bonding situation of the (Ng-H-Ng)^+^ and (Ng-H-Ng’)^+^. The Ng-H and Ng’-H contacts range, in fact, between strong covalent bonds to weak, non-covalent interactions, and their regular variability clearly illustrates the peculiar capability of the noble gases to undergo interactions covering the entire spectrum of the chemical bond.

## 1. Introduction

The recent detection of ArH^+^ and HeH^+^ in various galactic and extragalactic regions [1,2,3,4] is currently revitalizing interest in noble-gas ionic species [5], especially those conceivably occurring in outer space [6,7]. Particularly relevant in this regard are the triatomic (Ng-H-Ng)^+^ and (Ng-H-Ng’)^+^ (Ng, Ng’ = noble-gas atom). These prototypical proton-bound complexes, still elusive in the bulk phase, are best investigated under the isolated conditions of the gas phase [8,9,10,11,12,13,14,15,16,17,18,19,20,21,22,23,24,25,26] or in cold matrices [27,28,29,30,31,32,33,34]. Extensive theoretical information is also already available [35,36,37,38,39,40,41,42,43,44,45,46,47,48,49,50,51,52,53,54,55,56,57,58,59,60,61,62,63,64,65,66,67,68,69,70,71,72,73]. The first observed species were the homonuclear He_2_H^+^, Ne_2_H^+^, and Ar_2_H^+^, detected so far in the gas phase from ionized mixtures of Ng and H_2_ [8,9]. Their formation was ascribed to two ionic precursors, namely the NgH^+^ and Ng_2_^+^, and various subsequent studies [10,11,12,13,14,15,16] actually confirmed the efficiency of the reactions between He_2_^+^ and, especially, Ar_2_^+^ and H_2_. The helium cluster ions He*_n_*H^+^ (*n* ≤ 14), including He_2_H^+^, were also detected using ion sources operated at low temperature [17], and, more recently, the use of helium nanodroplets doped with Ng allowed the formation of a wide family of Ng*_n_*H^+^ (Ng = He, Ne, Ar, Kr), including the simplest Ng_2_H^+^ [18,19,20,21,22,23]. In any case, despite being observed since long ago, the gaseous Ng_2_H^+^ remained structurally unassigned until recently [24,25,26], when infrared (IR) spectroscopy revealed, in particular, the linear, centrosymmetric structure of (Ar-H-Ar)^+^ and (He-H-He)^+^. These findings actually confirmed the information already available from previous studies performed in cold matrices. Thus, nearly contemporary to the first reports about the ionized gaseous Ng/H_2_ mixtures [8,9], attention was paid to the IR absorptions of Ar/H_2_ or Kr/H_2_matrix samples deposited after the gas mixture was passed through a glow discharge [27]. Lines were observed, in particular, at 905 and 852 cm^−1^ (shifted to 644 and 607 cm^−1^, respectively, in experiments performed with Ar/D_2_ and Kr/D_2_), and assigned to H atoms trapped in the interstitial sites of Ar and Kr matrices. A different explanation was, however, soon after proposed [28,29], the carriers of the bands at 905/644 cm^−1^ being identified as the ionic Ar*_n_*H^+^/Ar*_n_*D^+^, in which *n* remained unspecified. Definitive evidence in this regard was obtained ten years later [30,31], when the bands tentatively assigned to the Ng*_n_*H^+^ (Ng = Ar, Kr, Xe) were unambiguously identified as the stretching absorptions of the triatomic (Ng-H-Ng)^+^, appearing in the IR spectra as a (*ν*_3_ + *nν*_1_) progression of the symmetric (*ν*_1_) and antisymmetric (*ν*_3_) motions, with *n* up to 4 for (Xe-H-Xe)^+^. Subsequent experiments confirmed these assignments, showing also a dependence of the corresponding wavenumbers on the nature of the trapping matrix [32,33,34].

To summarize, all five homonuclear Ng_2_H ^+^ (Ng = He-Xe) are well established species, experimentally observed in the gas phase and/or in solid matrices, and confidently assigned (with the sole exception of Ne_2_H^+^) as (Ng-H-Ng)^+^ by IR spectroscopy. The structural compactness suggested by the experiments is also consistent with the theoretical results [35,36,37,38,39,40,41,42,43,44,45,46,47,48,49,50,51,52,53,54,55,56,57,58], unraveling short Ng-H bonds of covalent character [59,60], and appreciable thermochemical stability with respect to the loss of Ng(s). When going to the heteronuclear (Ng-H-Ng’)^+^, the situation changes even appreciably. The numerous calculations already available [33,34,43,55,61,62,63,64,65,66,67,68,69,70,71,72,73] unravel, in fact, structurally asymmetric species, the (formal) H^+^ being closer and more tightly bound to the atom having the higher proton affinity (PA). The difference between the PA of Ng and Ng’ may actually arrive up to ca. 77 kcal mol^−1^ [74], and this produces variable effects on the geometry, thermochemical stability, and IR absorptions of the various (Ng-H-Ng’)^+^ [33,43,61,62,63,65]. The bonding character of these ions features also intriguing variabilities [63], but a quantitative and accurate analysis of their stabilizing interactions is, essentially, still missing. This is the major issue addressed in the present study, performed by the method that we recently proposed to analyze the chemical bonds involving the noble-gas atoms [75,76,77]. The obtained results actually unraveled a richness of bonding motifs, ranging from covalent bonds of different strength to weak non-covalent interactions. The bonding situations resulted also strictly related to the structure and stability of the investigated ions, and these relationships are also examined and discussed.

## 2. Method of Bonding Analysis

The method that we recently proposed to analyze the bonding character of noble-gas compounds is extensively discussed in key references [75,76,77] so we briefly recall here only the most salient features.

Our taken approach relies on the examination of the plotted shape of the local electron energy density *H*(***r***) [75,78,79], and on the values that this function takes over the volume *Ω*_s_ enclosed by the *s*(***r***) = 0.4 reduced density gradient (RDG) isosurface [80,81] associated with the bond critical point (BCP) that is located for a given Ng-X bond (X = binding partner) from the topological analysis of the electron density *ρ*(***r***) [82]. Ancillary indices include the size of *Ω*_s_, the total electronic charge enclosed by *Ω*_s_, *N*(*Ω*_s_), the average electron density over *Ω*_s_, *ρ*_s_(ave) = *N*(*Ω*_s_)/*Ω*_s_, and the average, maximum, and minimum value of *H*(***r***) over *Ω*_s_, *H_s_*(ave, max, min). As discussed previously [75], the *H*(***r***) partitions the atomic space in two well-recognizable regions, namely an inner one of negative values, indicated as *H*^−^(***r***), and an outer one of positive values, indicated as *H*^+^(***r***). The boundary of these regions falls at a distance *R*^−^, that is typical of each atom; at this distance, *H*(***r*** = *R*^−^) = 0. Interestingly, when two atoms form a chemical bond, their *H*^−^(***r***) and *H*^+^(***r***) regions combine in modes that signal the nature of the interaction. Particularly for the Ng-X bonds, it is possible to distinguish three types of interactions, indicated as A, B, or C. In interactions of type A, the atoms overlap all the contour lines of their *H*^+^(***r***) regions, and part of the contour lines of their inner *H*^−^(***r***) regions, the bond appearing as a continuous region of negative values of *H*(***r***), plunged in a zone of positive values. The interaction is topologically-signed by a (3 + 1) critical point of the *H*(***r***) (denoted as the HCP) falling on the bond axis. In interactions of type B, the *H*^−^(***r***) region of Ng is, again, overlapped with the *H*^−^(***r***) region of the binding partner, but (*i*) no HCP does exist on the bond axis, and (*ii*) the Ng-X inter-nuclear region does include a (more or less wide) region of positive *H*(***r***). In interactions of type C, the Ng and the binding partner overlap only part of their *H*^+^(***r***) regions, their *H*^−^(***r***) regions remaining, instead, perfectly closed, and separated by a (more or less wide) region of positive *H*(***r***). The bond thus appears as two clearly distinguishable *H*^−^(***r***) regions, separated by a region of positive values of *H*(***r***).

Any Ng-X is assigned as covalent (Cov) if (*i*) it is of type A, and (*ii*) the electron density at the BCP, *ρ*(BCP) is at least 0.08 *ea*_0_^−3^. The strength of a Cov bond is also quantified in terms of the bond degree (BD). Borrowing a concept introduced so far by Espinosa et al. [83], this index is defined by the equation:(1)$$\mathrm{BD}\left(\mathrm{Cov}\right)=-\frac{H\left(\mathrm{HCP}\right)}{\rho \left(\mathrm{HCP}\right)}$$
where *H*(HCP) and *ρ*(HCP) are, respectively, the *H*(***r***) and the *ρ*(***r***) at the HCP of the Ng-X bond.

Any Ng-X not fulfilling the criteria of covalency [i.e., it is of type B or C, or, if of type A, its *ρ*(BCP) is lower than 0.08 *e a*_0_^−3^] is further assayed by integrating the *H*(***r***) over *Ω*_s_. If the function is, invariably, positive over the entire volume, the interaction is assigned as non-covalent (nCov). If the function is partially or fully negative, the Ng-X is assigned as partially-covalent (pCov), and distinguished as H^+/−^, H^−/+^, and H^−^, the superscript indicating that, over *Ω*_s_, the *H*(***r***) is ranging from negative to positive, but, on the average, it is positive (H^+/−^) or negative (H^−/+^), or that it is, invariably, negative (H^−^). Interactions of type C are also assayed in terms of the degree of polarization of Ng, DoP(Ng), an index that measures, in essence, the deformation of the *H*^−^(***r***) region of Ng arising from the interaction with X [76]. It is, in particular, defined by the equation:(2)$$\mathrm{DoP}\left(\mathrm{Ng}\right)=\frac{[R_{\mathrm{Ng}}^{-}\left(\mathrm{Ng}-X\right)-R_{\mathrm{Ng}}^{-}]\times 100}{R_{\mathrm{Ng}}^{-}}$$
where $R_{\mathrm{Ng}}^{-}\left(\mathrm{Ng}-X\right)$ is the radius of the *H*^−^(***r***) region of Ng along the axis formed by Ng and the Ng-X BCP, and $R_{\mathrm{Ng}}^{-}$ is the radius of the *H*^−^(***r***) region of the free atom. Numerous illustrative examples of bonds classification are given in [77].

## 3. Computational Details

The calculations were performed at the coupled cluster level of theory, with inclusion of single and double substitutions, and an estimate of connected triples, CCSD(T) [84], using the Dunning’s correlation consistent aug-cc-pV*n*Zbasis sets (*n* = T, Q, 5) [85] (henceforth denoted as aV*n*Z). The Xe atom was treated by the Stuttgart/Cologne small-core (28 electrons), scalar-relativistic effective core potential (ECP-28) [86], and the jointly-designed aV*n*Z-PP basis sets. The calculations were performed by explicitly correlating the 2*s*^2^2*p*^6^ (Ne), 3*s*^2^3*p*^6^ (Ar), 4*s*^2^4*p*^6^ (Kr), and 5*s*^2^5*p*^6^ (Xe) outer electrons. The CCSD(T) correlation energies were extrapolated to the complete basis set (CBS) limit using the cubic extrapolation formula [87]:(3)$$E^{corr}\left(\mathrm{CBS}\right)=\frac{E_{5}^{corr}\times 5^{3}-E_{4}^{corr}\times 4^{3}}{5^{3}-4^{3}}$$
where $E_{4}^{corr}$ and $E_{5}^{corr}$ are, respectively, the CCSD(T)/aVQZ and CCSD(T)/aV5Z correlation energies. All the CCSD(T) calculations were performed with the CFOUR program (V2.1) [88].

The bonding analysis was accomplished at the CCSD(T)/aVTZ level of theory, the main analyzed functions being the *ρ*(***r***) [82], the *H*(***r***) [75,78,79], and the RDG, and itsrelated non-covalent interactions (NCI) indices [80,81]. The *ρ*(***r***) is defined by the equation [82]:(4)$$\rho \left(r\right)=\overset{}{\underset{i}{\sum }}\eta_{i}{\left|\varphi_{i}\left(r\right)\right|}^{2}$$
where $\eta_{i}$ is the occupation number of the natural orbital $\varphi_{i}$, in turn expanded as a linear combination of the basis functions.

The *H*(***r***) is the sum of the kinetic energy density *G*(***r***) and the potential energy density *V*(***r***):(5)H(r)=G(r)+ V(r)
The presently-employed definition [82,89] of the *G*(***r***) is given by the equation:(6)$$G\left(r\right)=\frac{1}{2}\overset{}{\underset{i=1}{\sum }}\eta_{i}{\left|\nabla \varphi_{i}\left(r\right)\right|}^{2}$$
where the sum runs over all the occupied natural orbitals $\varphi_{i}$ of occupation numbers $\eta_{i}$. The potential energy density *V*(***r***) is evaluated [82] from the local form of the virial theorem:(7)$$V\left(r\right)=\frac{1}{4}\nabla^{2}\rho \left(r\right)-2G\left(r\right)$$

The RDG is defined by the equation [80,81]:(8)$$s\left(r\right)=\frac{\left|\nabla \rho \left(r\right)\right|}{2{\left(3\pi^{2}\right)}^{\frac{1}{3}}\times \rho {\left(r\right)}^{\frac{4}{3}}}$$

Low-value *s*(***r***) isosurfaces (typically 0.3–0.6) appear among atoms undergoing any type of interaction, the NCIs emerging, in particular, by considering the spatial regions of low *ρ*(***r***). The low-*s*(***r***)/low-*ρ*(***r***) isosurfaces are, in turn, mapped in terms of the sign (*λ*_2_) × *ρ*(***r***), *λ*_2_ being the second eigenvalue (*λ*_1_ < *λ*_2_ < *λ*_3_) of the Hessian matrix of *ρ*(***r***). In essence, the sign of *λ*_2_ is used to distinguish between attractive (*λ*_2_ < 0) and repulsive (*λ*_2_ > 0) interactions, and the value of *ρ*(***r***) is exploited to rank the corresponding strength. In the present study, we calculated also the integral of a given property *P* [particularly the *ρ*(***r***) and the *H*(***r***)] over the volume *Ω*_s_ enclosed by the *s*(***r***) = 0.4 isosurface (s) at around the BCP located on any Ng-H bond path, *P*(*Ω*_s_). This integration was accomplished by producing an orthogonal grid of points that encloses the isosurface and applying the formula:(9)$$P\left(\Omega_{s}\right)=\underset{i\left(\mathrm{RDG}\;<s\right)}{\sum }P\left(r_{i}\right)d_{x}d_{y}d_{z}$$
where *P*(***r***_i_) is the value of *P* at the grid point ***r***_i_, and *d_x_*, *d_y_*, and *d_z_* are the grid step sizes in the *x*, *y*, and *z* directions, respectively (*d_x_* = *d_y_* = *d_z_* = 0.025 *a*_0_). The summation is carried out on all grid points ***r***_i_ having RDG <s.

The *ρ*(***r***), the *H*(***r***), and the *s*(***r***) were analyzed with the Multiwfn program (version 3.7.dev) [90] using the wave functions stored in the molden files generated with CFOUR, and properly formatted with the Molden2AIM utility program [91]. The two-(2D) plots of the *H*(***r***) were as well produced with Multiwfn, and include the standard contour lines belonging to the patterns ±*k* × 10*^n^* (*k* = 1, 2, 4, 8; *n* = −5 ÷ 6), together with the contour lines corresponding to the critical points specifically located from the topological analysis of the *H*(***r***).

## 4. Results and Discussion

The presently-investigated species include the five homonuclear (Ng-H-Ng)^+^ and the ten heteronuclear (Ng-H-Ng’)^+^. All these ions possess linear structures, and their CCSD(T)/aVTZ bond distances and harmonic vibrational frequencies are shown in Figure 1 and quoted in Table 1. Table 2 shows the values of Δ*R*(Ng-H), namely the elongation with respect to the diatomic NgH^+^ of the Ng-H bond of the various (Ng-H-Ng’)^+^.

The thermochemical stabilities of the investigated ions were assessed by computing the CCSD(T)/CBS energy change of the two-body (2B) dissociations:(Ng-H-Ng’)^+^ → NgH^+^/Ng’H^+^ + Ng’/Ng (10)
distinguished here using the notations 2B(Ng) and 2B(Ng’), and the three-body (3B) dissociation:(Ng-H-Ng’)^+^ → Ng + Ng’ + H^+^(11)

The obtained values are quoted in Table 1. The results of the CCSD(T)/aVTZ bonding analysis are given in Figure 2 and in Table 3.

### 4.1. The NgH^+^ and the Homonuclear (Ng-H-Ng)^+^ (Ng = He-Xe)

The diatomic NgH^+^ are typically covalent species. As shown in Figure 2 and Table 3, their plotted *H*(***r***) is of type A, and their *ρ*(BCP) is definitely higher than 0.08 e a_0_^−3^. The values of BD, ranging between 1.217 hartree *e*^−1^ (HeH^+^) and 0.824 hartree *e*^−1^ (XeH^+^), are also relatively high, and positively-correlated with rather high stretching frequencies ranging between 3206 cm^−1^ (HeH^+^) and 2297 cm^−1^ (XeH^+^). Compared with the NgH^+^, the homonuclear (Ng-H-Ng)^+^ are structurally less compact and thermochemically less stable. Thus, the Ng-H distance is invariably longer, Δ*R*(Ng-H) being computed as ca. 0.15 Ǻ for Ng = He and Ne, and ca. 0.22–0.26 Ǻ for Ng = Ar, Kr, and Xe, and the *ν*_3_ stretching frequency reduces in the range between 1616 cm^−1^ (Ne-H-Ne)^+^ and 846 cm^−1^ (Xe-H-Xe)^+^. In addition, the half of the energy change of the 3B dissociation [Equation (11)], representing, in practice, the average proton affinity (aPA) of two Ng atoms sharing the (formal) H^+^, is, invariably, lower than the PA of Ng, namely the energy change of the reaction:NgH ^+^ → Ng + H^+^(12)

Based on the data quoted in Table 1, the difference between PA(Ng) and aPA(Ng), 17.0 kcal mol^−1^ for Ng = He, 20.5 kcal mol^−1^ for Ng = Ne, 45.8 kcal mol^−1^ for Ng = Ar, 51.9 kcal mol^−1^ for Ng = Kr, and 59.8 kcal mol^−1^ for Ng = Xe, unravel, in particular, a thermochemical weakening effect that progressively increases when going from He to Xe. In any case, the bonds of any (Ng-H-Ng)^+^ are still assigned as Cov. As shown in Figure 2 and Table 3, the plotted *H*(***r***) is, invariably, of type A, and the *ρ*(BCP) is, invariably, higher than 0.08 *ea*_0_^−3^. The BD is, however, lower, than that of the corresponding NgH^+^, with percentage decreases of ca. 23% for Ng = Ne, and ca. 33-40% for Ng = He, Ar, Kr, and Xe. In this regard, it is of interest to note that, when going from H_2_(*R* = 0.743 Å) to the linear centrosymmetric (H-H-H)^−^ (*R* = 1.059 Å), isoelectronic with (He-H-He)^+^, the BD of the H-H bond decreases as well by ca. 37% (1.150 vs. 0.722 hartree *e*^−1^). However, at variance with (He-H-He)^+^, the covalent H_3_^−^ is a first-order saddle point on the potential energy surface, connecting two equivalent linear van der Waals energy minima H^−^(H_2_), and less stable than H^−^ + H_2_ by ca. 10 kcal mol^−1^ [92,93,94]. Major differences referable to the positive vs. negative charge of the two ions.

To summarize, when going from NgH^+^ to (Ng-H-Ng)^+^, the strength of the Ng-H interaction decreases, but its character does not change. In the heteronuclear (Ng-H-Ng’)^+^, the mutual effects of Ng/Ng’ on the adjacent Ng’-H/Ng-H bond are, instead, more complex and variegated. This is best discussed in the subsequent paragraph.

### 4.2. The Heteronuclear (Ng-H-Ng’)^+^ (Ng = He-Xe)

The structure, stability, and bonding character of the heteronuclear (Ng-H-Ng’)^+^ depend, essentially, on the difference between the PA of Ng and Ng’. The PAs of He and Ne are comparable (47.1 vs. 52.8 kcal mol^−1^, see Table 1) and, in fact, in the (He-H-Ne)^+^ the (formal) H^+^ is, essentially, shared between the two atoms. The He-H and Ne-H distances are comparable (0.9589 vs. 1.1082 Ǻ), the 2B(Ne) channel is only slightly more endothermic than the 2B(He) (17.6 vs. 11.9 kcal mol^−1^), and the endothermicity of the 3B channel, 64.7 kcal mol^−1^, is definitely lower than the sum of the PA of He and Ne, 99.9 kcal mol^−1^, thus confirming a mutual weakening effect. The values of Δ*R*(He-H) and Δ*R*(Ne-H) are also comparable, and predicted as ca. 0.18 and ca. 0.12 Ǻ, respectively. Consistently, the bonding situation of the (He-H-Ne)^+^ is similar to that occurring in the homonuclear (Ng-H-Ng)^+^, and accounted by the two nearly-equivalent resonance structures (He-H^+^)(Ne) and (He)(H-Ne^+^). Both the He-H and Ne-H bonds are, in fact, assigned as Cov, with a BD of 0.650 and 0.898, respectively. When going from He and Ne to Ar, the PA appreciably increases up to 93.6 kcal mol^−1^. Consistently, (He-H-Ar)^+^ and (Ne-H-Ar)^+^ are best described, respectively, by the resonance structures (He)(H-Ar^+^) and (Ne)(H-Ar^+^), accounting for weakly-bound complexes of He and Ne with the covalent Ar-H^+^. The He-H/Ne-H distances are, in fact, as long as 1.5157/1.5790 Ǻ, the 2B(He)/2B(Ne) channel is endothermic by only 2.08/3.83 kcal mol^−1^, and the *ν*_3_ stretching frequencies of 2569 and 2399 cm^−1^, respectively, are lower than, but comparable with the stretching absorption of ArH^+^, 2729 cm^−1^. These structural assignments anticipate the results of the bonding analysis. Thus, as shown in Figure 2, the plotted H(**r**) of both (He-H-Ar)^+^ and (Ne-H-Ar)^+^ clearly shows the H^−^(**r**) regions of Ng and ArH^+^ that are just touching (Ng = Ne) or even separate(Ng = He). The adjacent He or Ne only slightly perturb the Ar-H bond, actually assigned as Cov, with a BD of 0.964 hartree e^−1^ (Ng = He) and 0.944 hartree e^−1^ (Ng = Ne) that is only slightly lower than the BD of ArH^+^, 0.976 hartree e^−1^ (see Table 3). The He-H and Ne-H bonds are, instead, definitely weaker than those occurring in the HeH^+^ and NeH^+^, even though they still maintain a contribution of covalency. They are, in fact, assigned as pCov(C/H^−^) and pCov(A/H^−^), respectively, with corresponding *ρ*_s_(ave) of 0.0249 and 0.0383 *e a*_0_^−3^, and *H*_s_(ave) of −0.0024 and −0.0062 hartree *a*_0_^−3^, respectively. The PA of Kr (105.0 kcal mol^−1^) and Xe (120.2 kcal mol^−1^) is definitely higher than that of He and Ne, and in fact, in the four complexes (Ng-H-Kr)^+^ and (Ng-H-Xe)^+^ (Ng = He, Ne), the (formal) H^+^ is definitely closer to Kr or Xe, with Kr-H or Xe-H distances that are only slightly elongated with respect to the diatomic KrH^+^ or XeH^+^ [the Δ*R*(Kr-H) and Δ*R*(Xe-H) are, invariably, less than 0.01 Ǻ, see Table 2]. Consistently, as shown in Table 1, the 2B(He) and 2B(Ne) channels are endothermic by only 0.6-2.4 kcal mol^−1^, and the endothermicity of the 3B channels are, invariably, nearly coincident with the PA of Kr or Xe. The *ν*_3_ stretching frequencies of (He-H-Kr)^+^, 2482 cm^−1^, (Ne-H-Kr)^+^, 2408 cm^−1^, (He-H-Xe)^+^, 2291 cm^−1^, and (Ne-H-Xe)^+^, 2271 cm^−1^, are also close to the stretching absorptions of KrH^+^, 2528 cm^−1^, or XeH^+^, 2297 cm^−1^. In essence, these systems should be considered as a diatomic ion separated from a nearly isolated Ng atom. Consistently, the plots of Figure 2 and the data quoted in Table 3 unravel a clearly distinguishable covalent KrH^+^ or XeH^+^ moiety, whose BD is coincident or quite close to that of the diatomic ions, and whose *H*^−^(***r***) region is well separated from that of He or Ne. The three (He-HKr)^+^, (He-HXe)^+^, and (Ne-HXe)^+^ bonds are, indeed, all assigned as nCov, and their predicted *N*(*Ω*_s_) (less than 1 m*e*) and *ρ*_s_(ave) (less than 0.015 *e a*_0_^−3^) are typical of non-covalent contacts [77]. The computed DoP(He) and DoP(Ne) of 7.92, 4.49, and 3.22, respectively, are, however, rather large [76], and clearly signal He or Ne atoms that are appreciably polarized toward the adjacent cation. As a matter of fact, in the (Ne-H-Kr)^+^, the polarization of Ne (DoP = 4.53) is already sufficient to make the (Ne-HKr)^+^ bond partially covalent [pCov(C/H^+/−^)], with a *ρ*_s_(ave) of 0.0228 *e a*_0_^−3^.

The *ν*_3_ absorptions of (Ar-H-Kr)^+^, 1390 cm^−1^, (Ar-H-Xe)^+^, 1849 cm^−1^, and (Kr-H-Xe)^+^, 1410 cm^−1^, are significantly lower than the stretching frequencies of ArH^+^, KrH^+^, and XeH^+^. This resembles the situation occurring in (He-H-Ne)^+^, and in the five homonuclear (Ng-H-Ng)^+^ (Ng = He-Xe) (*vide supra*), and could suggest relatively similar bonding situations. As a matter of fact, the plotted *H*(***r***) of the (Ar-H-Kr)^+^, (Ar-H-Xe)^+^, and (Kr-H-Xe)^+^ (see Figure 3) invariably shows the continuing overlapping of the *H*^−^(***r***) regions that is typical of covalent or partially-covalent bonds. The quantitative analysis, however, unravels mutual weakening effects of Ng and Ng’ that are peculiar of the three ions. Thus, in the (Ar-H-Kr)^+^, the Δ*R*(Kr-H) amounts to only 0.1102 Ǻ, and the Kr-H bond is assigned as Cov, even though its BD of 0.782 hartree *e*^−1^ is lower than the BD of KrH^+^, 0.937 hartree *e*^−1^. The Δ*R*(Ar-H), 0.4129 Ǻ, is, instead, large enough to produce a Ar-H bond assigned as pCov(A,H^−^), even though still featuring a definitely negative *H*_s_(ave) of −0.0333 hartree *a*_0_^−3^, and a relatively high *ρ*_s_(ave) of 0.0715 *e a*_0_^−3^. In essence, the difference of 11.4 kcal mol^−1^ between the PA of Ar and Kr produces a resonance structure (Ar)(H-Kr^+^) that is more weighting than, but not by far prevailing on the (Ar-H^+^)(Kr). When going from (Ar-H-Kr)^+^ to (Ar-H-Xe)^+^, the Δ*R*(Ar-H) further increases up to 0.6701 Ǻ, and the endothermicity of the 2B(Ar) channel decreases from 10.4 to 5.70 kcal mol^−1^ (see Table 1). Consistently, the Ar-H bond, still assigned as pCov(A/H^−^), has a lower degree of covalency, signaled by an increased (less negative) *H*_s_(ave) of -0.0086 hartree *a*_0_^−3^, and a dereased *ρ*_s_(ave) of 0.0387 *e a*_0_^−3^. The resonance structure (Ar)(H-Xe^+^) is, thus, by far prevailing on the (Ar-H^+^)(Xe). The difference between the PA of Kr and Xe, 15.2 kcal mol^−1^, is only slightly larger than that between the PA of Ar and Kr, and the bonding situation of (Kr-H-Xe)^+^ is, indeed, similar to that occurring in (Ar-H-Kr)^+^. Thus, the Δ*R*(Xe-H) amounts to 0.0984 Ǻ, and the bond is assigned as Cov, with a BD of 0.709 hartree *e*^−1^. The Δ*R*(Kr-H) arrives, instead, up to 0.4998 Ǻ, and the Kr-H bond is, actually, assigned as pCov(A/H^−^), with a *H*_s_(ave) of −0.0192 hartree *a*_0_^−3^, and a *ρ*_s_(ave) of 0.0556 *e a*_0_^−3^. Consistently, the endothermicity of the 2B(Kr) channel is appreciably lower than that of the 2B(Xe) channel (8.68 vs. 23.9 kcal mol^−1^).

Overall, the structure, stability, and bonding character of the fifteen proton-bound (Ng-H-Ng)^+^ and (Ng-H-Ng’)^+^ are strictly related, their qualitative and also quantitative trends mirroring the periodically-variable mutual effects of Ng and Ng’. A summarizing overview of these relationships is given in the subsequent paragraph.

### 4.3. An Overview of the (Ng-H-Ng’)^+^ (Ng = Ng’ or Ng ≠ Ng’)

As already noted previously [63], an effective mode to overview the properties of the fifteen (Ng-H-Ng’)^+^ (Ng = Ng’ or Ng ≠ Ng’) is to analyze the effects produced on the diatomic NgH^+^ by the ligation with Ng’. The sharing of the (formal) H^+^ between Ng and Ng’ generally produces an elongation of the Ng-H bond, the effect depending on the resistance of Ng to donate the proton, and on the ability of Ng’ to accept it. These competing tendencies both increase by increasing the PA of Ng and Ng’, and, therefore, increase on going from He to Xe.Consistently, the values of Δ*R*(Ng-H) quoted in Table 2 clearly unravel i) the elongation of any Ng-H bond by any Ng’ ii) for any Ng’, a decreased elongation when going from He-H to Xe-H, and iii) for any Ng-H, an increased elongation when going from Ng’ = He to Ng’ = Xe. The quantitative effects of the nature of Ng and Ng’, and, in particular, of the values of their PA are best caught by inspecting the graphs given in Figure 3, showing the trends of the Δ*R*(Ng-H) as a function of Ng’.

Thus, the bond distances of HeH^+^ and NeH^+^, only slightly and comparably elongated by ligation with He or Ne (by ca. 0.12 and ca. 0.18 Ǻ), suddenly increase upon ligation with Ar (by ca. 0.6–0.7 Ǻ), and further increase nearly linearly by ligation with Kr (by ca. 0.8–1.0 Ǻ), and Xe (by ca. 1.0 and 1.3 Ǻ). These structural effects clearly mirror the changes occurring in the thermochemical stability and bonding character of the (He-H-Ng’)^+^ and (Ne-H-Ng’)^+^. Thus, the endothermicty of the 2B(He) and 2B(Ne) channels, ranging between ca. 12 and ca. 18 kcal mol^−1^ for Ng’ = He or Ne, drastically reduces to ca. 2–4 kcal mol^−1^ for Ng’ = Ar, and arrives down to ca. 1 kcal mol^−1^ or even less for Ng’ = Kr or Xe. Meanwhile, as schematically shown in Figure 4, when going from Ng’ = He or Ne to Ng’ = Ar, the character of the (He-HNg’)^+^ and (Ne-HNg’)^+^ bonds features a major shift from the Cov to the pCov domain, and arrives up to the nCov for (He-HKr’)^+^, (He-HXe’)^+^, and (Ne-HXe’)^+^.

As shown in Figure 3, due to their lowest PA, both He and Ne are, essentially, unable to elongate the bond distances of ArH^+^, KrH^+^, and XeH^+^. Appreciable effects on these ions are, instead, exerted by Ar, and, especially, Kr and Xe. One also notes that, on going from Ng’ = Ne to Ng’ = Kr, the dependencies of Δ*R*(Ar-H), Δ*R*(Kr-H), and Δ*R*(Xe-H) are, essentially, linear but progressively less pending, and become slightly more pending on going to Ng’ = Xe. These trends mirror, in essence, the progressively-increased reluctance of ArH^+^, KrH^+^, and XeH^+^ to share the (formal) H^+^, and the highest ability of Xe to accept it. As a matter of fact, as shown in Figure 4, on going from the (Ne-H-Ng’)^+^ to the (Xe-H-Ng’)^+^, the pCov domain of the Ng-H bond becomes progressively less populated, and all the five (Xe-HNg’)^+^ bonds are, actually, assigned as Cov. In any case, as also eye-caught by inspecting the right part of Figure 4, the intrinsic strength of the various covalent Ng-H bonds progressively decreases on going from the He-H to the Xe-H; the complementary aspect of the progressively increased tendency of the Ng atoms to share their valence electrons when going from He to Xe.

## 5. Conclusions

One of the most fascinating aspects of the chemistry of the noble gases is their ability to form chemical bonds of quite diverse type, ranging from weak non-covalent contacts to strong covalent bonds. Exemplary in this regard are the presently-investigated proton- bound dimers (Ng-H-Ng)^+^ and (Ng-H-Ng’)^+^. The progressively-variable nature of the Ng-H and Ng’-H contacts parallels the concomitant variations of the bond distances and thermochemical stabilities, featuring as well regular trends as a function of Ng and Ng’. The structure, stability, and bonding character of these ionic complexes are, indeed, strictly related, and this suggests similar relationships for other groups of cationic noble-gas hydrides. Their further investigation is also stimulated by the current active interest for the conceivable role of various Ng*_m_*H*_n_*^+^ (*m*, *n* ≥ 1) in processes of astrochemical interest.

## Figures and Tables

**Figure 1 molecules-26-01305-f001:**
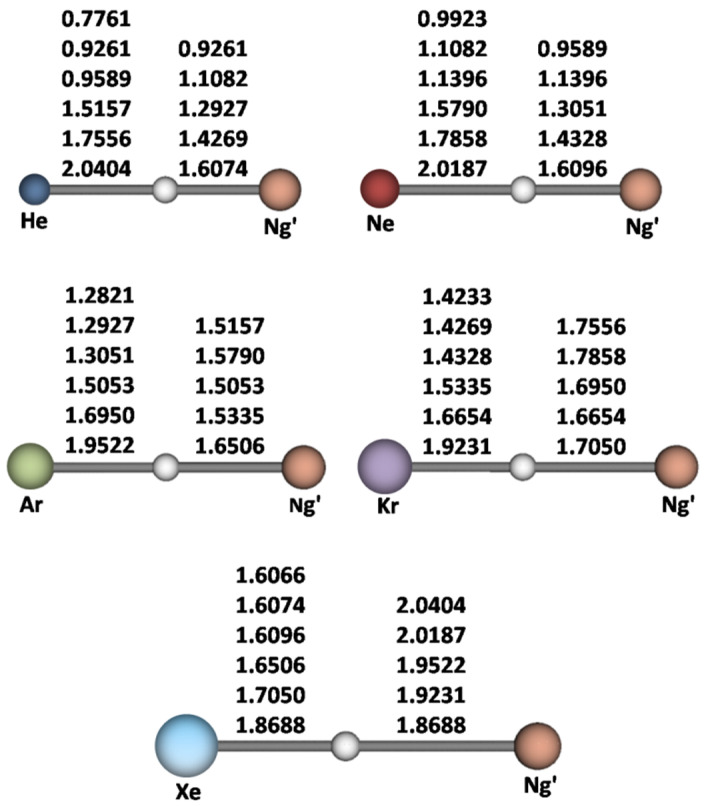
CCSD(T)/aVTZ bond distances (Å) of the (Ng-H-Ng’)^+^. For any Ng, from top to bottom, Ng’ = none, He, Ne, Ar, Kr, Xe.

**Figure 2 molecules-26-01305-f002:**
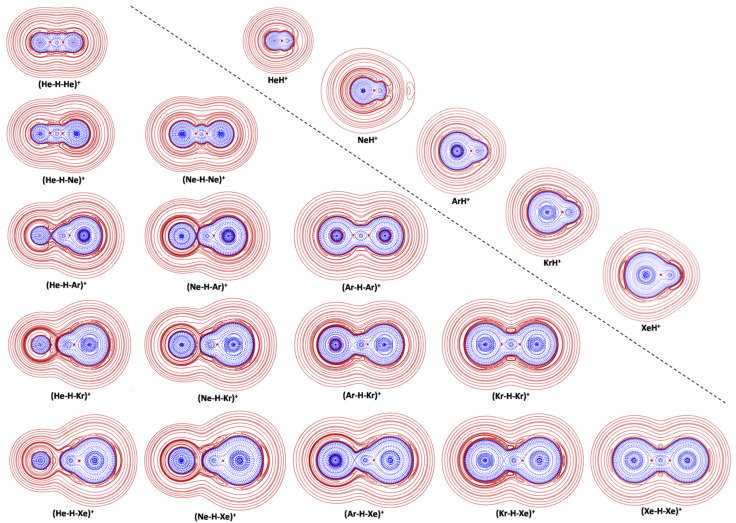
2D-plots of *H*(***r***) in the main plane of the NgH^+^ and (Ng-H-Ng’)^+^ (solid/brown and dashed/blue lines correspond, respectively, to positive and negative values). The red dots sign the HCP.

**Figure 3 molecules-26-01305-f003:**
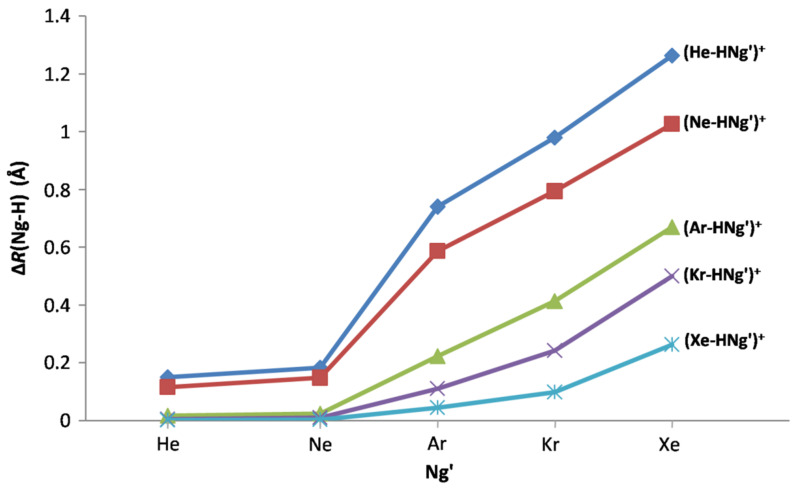
Elongation with respect to the diatomic NgH^+^, Δ*R*(Ng-H) (Ǻ), of the Ng-H bond distances of the (Ng-H-Ng’)^+^ as a function of Ng’.

**Figure 4 molecules-26-01305-f004:**
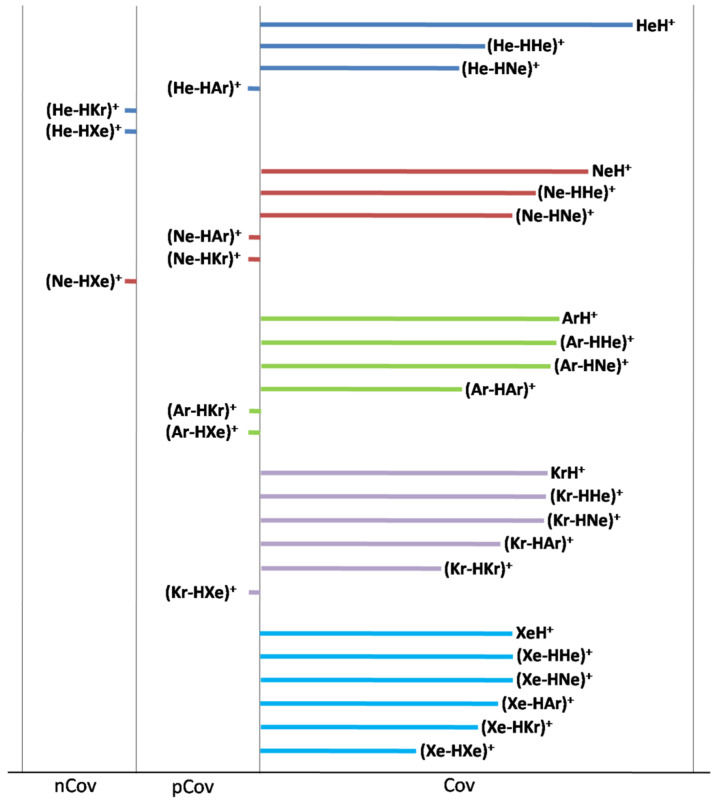
Character of the (Ng-HNg’)^+^ interactions. For the bonds of type Cov, the length of the bar is proportional to the BD.

**Table 1 molecules-26-01305-t001:** CCSD(T)/aVTZ bond distances (Å), harmonic vibrational frequencies (cm^−1^), and CCSD(T)/CBS dissociation energies (kcal mol^−1^) of the (Ng-H-Ng’)^+^ and NgH^+^.

(Ng-H-Ng’)^+^	*R*(Ng-H)	*R*(Ng’-H)	*ν* _1_ ^1^	*ν* _2_ ^2^	*ν* _3_ ^3^	Ng’H^+^ + Ng ^4^	NgH^+^ + Ng’ ^5^	Ng + Ng’ + H^+^ ^6^
(He-H-He)^+^	0.9261	0.9261	1138	958	1559	13.2	13.2	60.3
(He-H-Ne)^+^	0.9589	1.1082	853	882	1645	11.9	17.6	64.7
(He-H-Ar)^+^	1.5157	1.2927	234	333	2569	2.08	48.6	95.7
(He-H-Kr)^+^	1.7556	1.4269	159	193	2482	1.18	59.1	106.2
(He-H-Xe)^+^	2.0404	1.6074	109	102	2291	0.60	73.8	120.8
(Ne-H-Ne)^+^	1.1396	1.1396	519	850	1616	15.8	15.8	68.5
(Ne-H-Ar)^+^	1.5790	1.3051	169	431	2399	3.83	44.6	97.4
(Ne-H-Kr)^+^	1.7858	1.4328	110	268	2408	2.36	54.6	107.4
(Ne-H-Xe)^+^	2.0187	1.6096	81	188	2271	1.25	68.7	121.4
(Ar-H-Ar)^+^	1.5053	1.5053	322	705	956	15.5	15.5	109.0
(Ar-H-Kr)^+^	1.6950	1.5335	185	594	1390	10.4	21.8	115.4
(Ar-H-Xe)^+^	1.9522	1.6506	112	395	1849	5.70	32.4	125.9
(Kr-H-Kr)^+^	1.6654	1.6654	206	639	869	15.2	15.2	120.2
(Kr-H-Xe)^+^	1.9231	1.7050	113	506	1410	8.68	23.9	128.9
(Xe-H-Xe)^+^	1.8688	1.8688	150	575	846	14.4	14.4	134.6
HeH^+^	0.7761				3206			47.1
NeH^+^	0.9923				2946			52.8
ArH^+^	1.2821				2729			93.6
KrH^+^	1.4233				2528			105.0
XeH^+^	1.6066				2297			120.2

^1^ Σ symmetric stretching; ^2^ Π doubly-degenerate bending; ^3^ Σ asymmetric stretching; ^4^ 2B(Ng) channel; ^5^ 2B(Ng’) channel; ^6^ 3B channel.

**Table 2 molecules-26-01305-t002:** Elongation with respect to the diatomic NgH^+^, Δ*R*(Ng-H) (Å),of the CCSD(T)/aVTZ (Ng-HNg’)^+^ bond distances (Å).

Ng’	(He-HNg’)^+^	(Ne-HNg’)^+^	(Ar-HNg’)^+^	(Kr-HNg’)^+^	(Xe-HNg’)^+^
He	0.1500	0.1159	0.0106	0.0036	0.0008
Ne	0.1828	0.1473	0.0230	0.0095	0.0030
Ar	0.7396	0.5867	0.2232	0.1102	0.0440
Kr	0.9795	0.7935	0.4129	0.2421	0.0984
Xe	1.2643	1.0264	0.6701	0.4998	0.2622

**Table 3 molecules-26-01305-t003:** CCSD(T)/aVTZ type and properties of the Ng-H^+^, (Ng-HNg’)^+^ and (NgH-Ng’)^+^ bonds. For Cov bonds, the quoted *ρ*(BCP) (*e a*_0_^−3^), *ρ*(HCP) (*e a*_0_^−3^), *H*(HCP) (hartree *a*_0_^−3^), and BD (hartree *e*^−1^) are, respectively, the electron density at the BCP, and the electron density, the energy density, and the bond degree at the HCP. For pCov or nCov bonds, the quoted *Ω*_s_ (*a*_0_^3^),*N*(*Ω*_s_) (m*e*), *ρ*_s_(ave) (*e a*_0_^−3^), and *H_s_*(ave/max/min) (hartree *a*_0_^−3^) are, respectively, the volume enclosed by the *s*(***r***) = 0.4 isosurface at around the BCP, and the total electronic charge, the average electron density, and the average, maximum and minimum value of *H*(***r***) over *Ω*_s_.

		***ρ*(BCP)**	***ρ*(HCP)**	***H*(HCP)**	**BD**
He-H^+^	Cov	0.2057	0.2715	−0.3305	1.217
Ne-H^+^	Cov	0.2135	0.3083	−0.3293	1.068
Ar-H^+^	Cov	0.2324	0.2445	−0.2387	0.976
Kr-H^+^	Cov	0.2057	0.2079	−0.1948	0.937
Xe-H^+^	Cov	0.1677	0.1690	−0.1393	0.824
(He-HHe)^+^	Cov	0.1299	0.1547	−0.1136	0.734
(He-HNe)^+^	Cov	0.1187	0.1381	−0.0897	0.650
(HeH-Ne)^+^	Cov	0.1481	0.1812	−0.1627	0.898
(HeH-Ar)^+^	Cov	0.2246	0.2375	−0.2289	0.964
(HeH-Kr)^+^	Cov	0.2037	0.2063	−0.1924	0.933
(HeH-Xe)^+^	Cov	0.1677	0.1689	−0.1392	0.824
(Ne-HNe)^+^	Cov	0.1359	0.1602	−0.1317	0.822
(NeH-Ar)^+^	Cov	0.2163	0.2294	−0.2166	0.944
(NeH-Kr)^+^	Cov	0.2006	0.2034	−0.1881	0.925
(NeH-Xe)^+^	Cov	0.1671	0.1684	−0.1386	0.823
(Ar-HAr)^+^	Cov	0.1235	0.1325	−0.0870	0.657
(ArH-Kr)^+^	Cov	0.1538	0.1586	−0.1241	0.782
(ArH-Xe)^+^	Cov	0.1536	0.1554	−0.1206	0.776
(Kr-HKr)^+^	Cov	0.1103	0.1175	−0.0691	0.588
(KrH-Xe)^+^	Cov	0.1361	0.1387	−0.0983	0.709
(Xe-HXe)^+^	Cov	0.0940	0.0973	−0.0494	0.508
		***Ω*_s_**	***N*(*Ω*_s_)**	***ρ*_s_(ave)**	***H_s_*(ave/max/min)**
(He-HAr)^+^	pCov (C/H^−^)	0.0576	1.44	0.0249	−0.0024/−0.0012/−0.0043
(He-HKr)^+^	nCov (C)	0.0428	0.59	0.0138	0.0016/0.0020/0.0009
(He-HXe)^+^	nCov (C)	0.0320	0.23	0.0070	0.0021/0.0023/0.0020
(Ne-HAr)^+^	pCov (A/H^−^)	0.0831	3.18	0.0383	−0.0062/−0.0025/−0.0121
(Ne-HKr)^+^	pCov (C/H^+/−^)	0.0759	1.73	0.0228	0.0001/0.0025/−0.0029
(Ne-HXe)^+^	nCov	0.0651	0.87	0.0134	0.0018/0.0025/0.0012
(Ar-HKr)^+^	pCov (A/H^−^)	0.3511	25.1	0.0715	−0.0333/−0.0224/−0.0703
(Ar-HXe)^+^	pCov (A/H^−^)	0.2523	9.78	0.0387	−0.0086/−0.0047/−0.0173
(Kr-HXe)^+^	pCov (A/H^−^)	0.4932	27.4	0.0556	−0.0192/−0.0111/−0.0409

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
