# Peer review of "On the Proton-Bound Noble Gas Dimers (Ng-H-Ng)+ and (Ng-H-Ng’)+ (Ng, Ng’ = He-Xe): Relationships between Structure, Stability, and Bonding Character"

_molecules, 2021, doi:10.3390/molecules26051305_

Round 1

Reviewer 1 Report

This paper which theoretically reports on the structure, stability, and bonding character in some noble gas-H-noble gas [(Ng-H-Ng’)+] molecules is a good work which was competently carried out.  I recommend its publication in Molecules.  There are just a few minor matters that the authors should consider before publication proceeds.

a) The triatomic molecules which were studied are indeed isoelectronic to the linear (H3)- molecule. This should be mentioned. Moreover, it would be useful to compare the energetics computed for (Ng-H-Ng’)+ to those for (H3)- which could serve as a ‘reference’, in particular the dissociation energies and the bonding degree.

b) Some triatomic molecules such as (He-H-Xe)+ with a very small dissociation energy and a very long Ng-H distance (see Table 1) should rather be considered as a diatomic molecule separated from an isolated Ng atom.

c) Minor points

Table 3. What is the meaning of negative BD values?

Typo. P. 3, l. 107.  separated

Reviewer 2 Report

This work reports on the careful quantitative analysis of the stabilizing interactions in fifteen (Ng-H-Ng)+ and (Ng-H-Ng')+  noble-gas systems - prototypical proton-bound complexes - with Ng, Ng' = He-Xe. CCSD(T) method was used to estimate the interatomic distances, harmonic vibration frequencies and dissociation energies. 
Then the extended approach based on the QTAIM elements as electron density and the local electron energy density, etc. was used to characterise the specific atomic interactions.

The results support the well-known ability of the noble gases to form chemical bonds of  diverse type, from weak non-covalent interactions to strong covalent bonds. The Authors, taking the bond distances and thermochemical stability data in combination with electron density analysis, try to find the regular trends in the structure and properties of compounds studied.

This is an interesting work, which looks as an invitation to continue the investigation.

Specific comment: 
What does it mean (Table 3):  ρ(BCP) or  Ωs, ets? They are different dimension quantities!
